# Visualizing chest X-ray dataset biases using GANs

**Hao Liang**                                                    HL106@RICE.EDU
**Kangqi Ni**                                                    KN22@RICE.EDU
**Guha Balakrishnan**                                            GUHA@RICE.EDU
*Department of Electrical and Computer Engineering, Rice University, USA*

## Abstract

Recent work demonstrates that images from various chest X-ray datasets contain visual features that are strongly correlated with protected demographic attributes like race and gender. This finding raises issues of fairness, since some of these factors may be used by downstream algorithms for clinical predictions. In this work, we propose a framework, using generative adversarial networks (GANs), to visualize what features are most different between X-rays belonging to two demographic subgroups.

**Keywords:** Chest X-rays, fairness, bias, explainability, generative adversarial networks (GANs)

## 1. Introduction

Recent studies have demonstrated that patient bio-information like age, race, and gender are predictable from chest X-ray (CXR) images alone using deep learning models(Gichoya et al., 2022; Karargyris et al., 2019; Duffy et al., 2022). For example, in the "Reading Race" study, deep classifiers trained to predict race achieve 0.99 AUROC on several CXR datasets (Gichoya et al., 2022). This finding raises the question: "What visual cues discriminate different races?" Answering such a question can help mitigate potentially biased behavior of downstream algorithms that make decisions using this data. In this work, we propose a framework to visually explain the principal differences between different demographic subgroups in a medical imaging dataset. We first train an unconditional generative adversarial network (GAN) (Goodfellow et al., 2020; Liang et al., 2020; Lin et al., 2022) on the given image dataset. Next, we project the images onto the (trained) GAN's latent space and compute a direction in the latent space that differentiates a pair of classes (e.g., "Black" vs. "White" race groups). We traverse the latent space along that direction to produce image sequences that depict the main morphological and appearance changes in moving from one class to another.

There are related works that focus on visualizing subgroup differences associated with clinical attributes. One such study uses autoencoders (Cohen et al., 2021), which often produce blurry samples that do not clearly capture structural information. Others train *conditional* versions of GANs (Singla et al., 2023; Dravid et al., 2022), an expensive process since the GAN must be trained from scratch for each attribute of interest. In contrast to all these works, we demonstrate that deep generative models may be a useful tool to the medical imaging community to understand the biases within a medical imaging dataset.

## 2. Method

Our method consists of several components, visualized in Fig. 2 and described below.

**Generator training:** We train an unconditional StyleGAN2 generator (Karras et al., 2020a) $G(\cdot): R^d \to R^{H \times W \times 1}$, following the default training procedure introduced in that

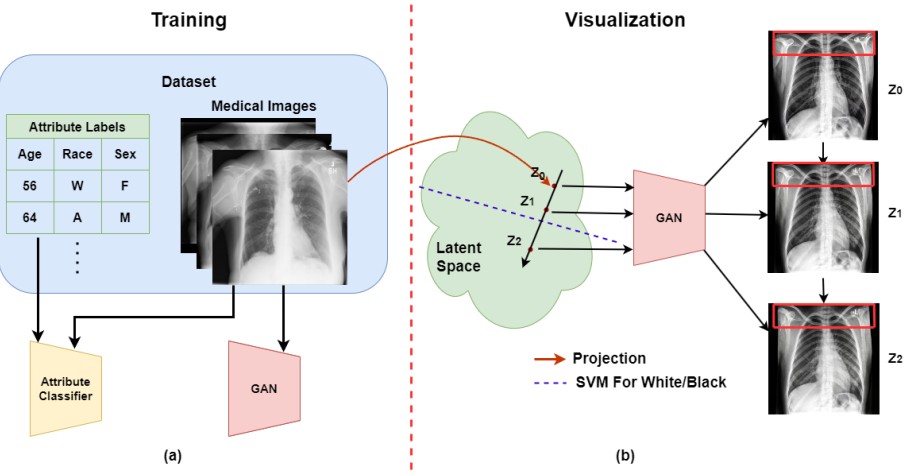

Figure 1: **Framework of our proposed method.** (a) We train a GAN on an image dataset, and a binary classifier on the images and labels for a demographic prediction task (e.g., White vs. Black race). (b) We project a subset of images onto the trained GAN's latent space. To ensure the projected images are reasonably reconstructed, we only keep projected images whose labels (predicted by the attribute classifier trained in (a)) agree with their original labels. We also fit an SVM hyperplane to separate the two classes in the latent space. Finally, we visualize the differences between the classes by starting at a latent code corresponding to a random image, and traversing along the normal direction of the SVM hyperplane, to generate a sequence of images showing a transformation.

paper. $d$ is the dimension of the "latent space" of the generator, and $H$ and $W$ are the height and width of the generated CXR. In our experiments, we trained $G(\cdot)$ on Chexpert (Irvin et al., 2019), a large public dataset containing $224,316$ CXRs. We only used frontal views, yielding $164,548$ CXRs. The training procedure takes roughly 24 hours on two Nvidia A100 GPUs.

**Attribute classifier training:** We train a separate deep attribute classifier $C(\cdot)$ : $R^{H \times W \times 1} \to R^1$ for each per-image binary attribute provided in the dataset. For multi-class labels such as race, we train a separate binary classifier for each pair of races.

**Image projection/SVM training:** Next, we follow the process introduced in (Karras et al., 2020b) to project a subset of CXR images $\{X_i\}_{i=1}^N$ onto $G$'s latent space, yielding latent codes $\{z_i\}_{i=1}^N$. We only retain those projected images whose labels (predicted by $C$) are the same as the original labels $\{L_i\}_{i=1}^N$, i.e., $C(G(z_i)) = L_i$. We then train a linear SVM to predict $L_i$ from $z_i$.

**Image sequence generation:** The normal vector $v$ of the trained SVM's hyperplane identifies the direction that best differentiates the two classes. We will use this fact to generate image sequences depicting the principal perceptual changes needed to convert a CXR belonging to one demographic class to another. In particular, we select the latent vector corresponding to a random dataset CXR, and move towards the opposite class in latent space in the direction of $v$. We concatenate images generated by intermediate latent codes along this traversal to produce a sequence.

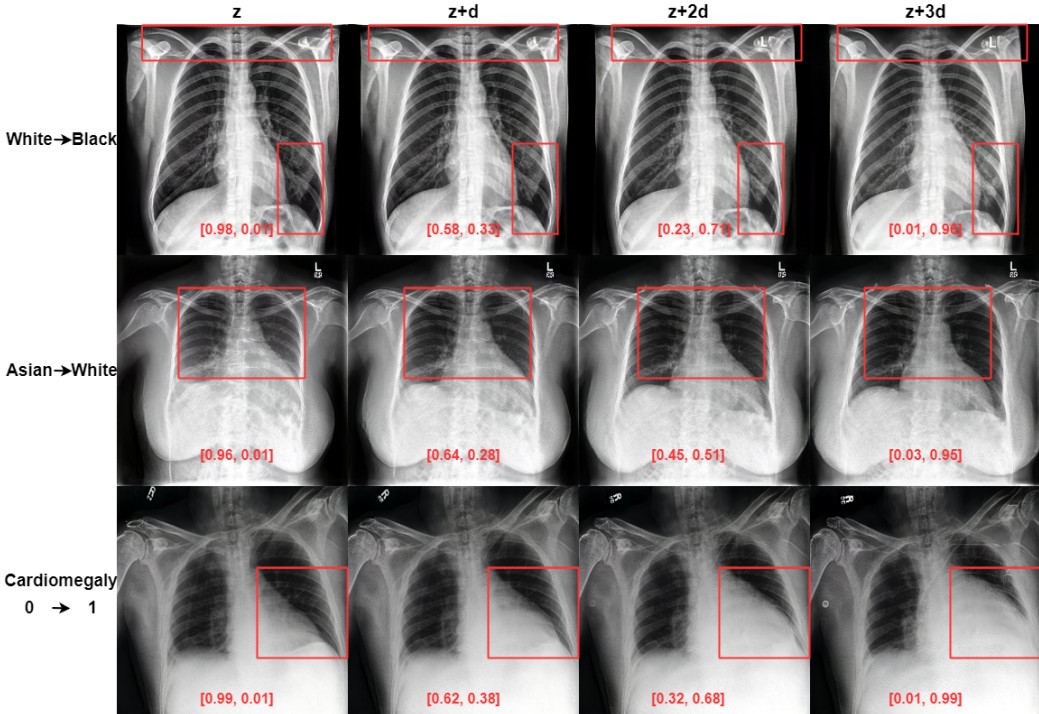

Figure 2: **Sample visualization results.** The left column corresponds to the projected initial image and the last three columns show images generated at different traversal distances in the latent space. The red text indicates the output probabilities predicted by the attribute classifier for each class. For example, the top left [0.98, 0.01] indicate the CXR has a 98% possibility of being white and 1% possibility of being black. We also use red boxes to highlight the areas that visually vary the most. For White/Black, the shoulder bone and right lung structures change shape, and the lungs become more opaque. For Asian/White, the entire chest shape changes and grows larger. These visualizations also explain why the Reading Race study (Gichoya et al., 2022) did not find race prediction to significantly change when blocking local regions. The proposed applied to *Cardiomegaly* enlarges the heart, in agreement with the known effect of that disease.

## 3. Results and discussion

We demonstrate our framework on ChexPert with *race* as the target attribute. We also validate our approach on the clinical attribute *Cardiomegaly*, which induces a known physiological change (enlarged heart). Sample results are shown and explained in Fig. 2.

**Conclusion** Our results show that an unconditional generative adversarial network can be a useful tool for visualizing differences between demographic groups of a CXR dataset. Our framework is fast and flexible, and can be applied to any binary attribute labels in the dataset. Future work includes analyzing generated sequences to thoroughly investigate demographic differences, and comparing results across different generative models.

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
