# OpenReview forum: "Visualizing chest X-ray dataset biases using GANs"
_MIDL.io/2023/Short_Paper_Track — MIDL 2023 Short paper track Poster_

### Official Review · Reviewer_fohx · 2023-04-17
**Interesting findings**

**Rating:** 6
**Confidence:** 4

**Review:**

This work presents a novel approach that highlights the most different features between different scanners that belong to different demographic groups.

Strengths:

- This method can be useful in practice, as it can potentially mitigate biases on downstream tasks that make decisions using data that contains multiple demographic groups.

- The abstract is easy to follow.

Weaknesses:

- Authors show only a few sample results to demonstrate that the proposed method works. I wonder whether a quantitative evaluation can be done to estimate how this method works (and which can allow future research to compare with it).

- There is no comparison to existing work, for example the ones cited that focus on visualizing subgroup differences associated with clinical attributes.

---

### Official Review · Reviewer_1XHP · 2023-04-22
**Visualizing chest X-ray dataset biases using GANs**

**Rating:** 5
**Confidence:** 5

**Review:**

This paper proposed a framework, using generative adversarial networks (GANs), to visualize what features are most different between X-rays belonging to two demographic subgroups. It's interesting to see such studies, which explore the potential biases in deep learning models. However, there are other explanable AI methods such as CAM, LIME, etc. A comparison or discussion with such methods would strengthen the contribution.